# Hormone-Glutamine Metabolism: A Critical Regulatory Axis in Endocrine-Related Cancers

**DOI:** 10.3390/ijms231710086

**Published:** 2022-09-03

**Authors:** Fengyuan Xu, Jialu Shi, Xueyun Qin, Zimeng Zheng, Min Chen, Zhi Lin, Jiangfeng Ye, Mingqing Li

**Affiliations:** 1Department of Bioengineering, University of Pennsylvania, Philadelphia, PA 19104, USA; 2Department of Gynecology, Hospital of Obstetrics and Gynecology, Shanghai Medical School, Fudan University, Shanghai 200010, China; 3Laboratory for Reproductive Immunology, Hospital of Obstetrics and Gynecology, Shanghai Medical School, Fudan University, Shanghai 200080, China; 4Institute for Molecular and Cell Biology, Agency for Science, Technology and Research, Singapore 138632, Singapore; 5NHC Key Lab. of Reproduction Regulation, Shanghai Institute for Biomedical and Pharmaceutical Technologies, Fudan University, Shanghai 201203, China

**Keywords:** glutamine, hormone, estrogen, progesterone, androgen, prostaglandin, thyroid hormone, insulin

## Abstract

The endocrine-related cancers and hormones are undoubtedly highly interconnected. How hormones support or repress tumor induction and progression has been extensively profiled. Furthermore, advances in understanding the role of glutamine metabolism in mediating tumorigenesis and development, coupled with these in-depth studies on hormone (e.g., estrogen, progesterone, androgen, prostaglandin, thyroid hormone, and insulin) regulation of glutamine metabolism, have led us to think about the relationship between these three factors, which remains to be elucidated. Accordingly, in this review, we present an updated overview of glutamine metabolism traits and its influence on endocrine oncology, as well as its upstream hormonal regulation. More importantly, this hormone/glutamine metabolism axis may help in the discovery of novel therapeutic strategies for endocrine-related cancer.

## 1. Introduction

Endocrine-related cancers generally indicate sex steroid-responsive cancers, including breast cancers, endometrial cancers, prostate cancers, and testis cancers, but there are other cancers, like thyroid and ovary cancers, which are also responsive to sex hormones [1,2,3]. Indeed, they are unique as most are hormone-dependent or produce hormones. Hormones, such as estrogen and androgen, play a pivotal role in these tumors, promoting carcinogenesis, angiogenesis, and invasiveness, as well as apoptosis [4,5,6]. For example, estrogen functions as the promoter of breast cancer and ovarian cancer through the ligand-activated transcription factor ER on the cell nucleus, and it can also stimulate endometrial glands and stromal cells to grow and proliferate during the menstrual cycle. DNA damage and mutations accumulate and increase the possibility of breast and ovarian cancer occurrence due to the imbalance between estrogen and progesterone during menstrual cycles. Meanwhile, prolonged exposure to estrogen is the leading cause of type I endometrial cancer [7,8,9], which usually expresses a high level of estrogen receptors (ER). Moreover, there is an interesting phenomenon whereby ERα activation may induce thyroid cancer, while wild-type ERβ (ERβ1) has protective effects against thyroid cancer [10]. Androgen and its receptors are reported to be associated with the carcinogenesis of the testes and prostate [11,12,13]. The imbalance between sex hormones and hormone receptors is the leading cause of testis cancer due to their considerable roles in normal organogenesis and spermatogenesis in testes and development in prostate [6,14]. Meanwhile, thyroid hormones triiodothyronine (T3) and thyroxine (T4) show effects on cancer proliferation, apoptosis, invasiveness, and angiogenesis through activation of the plasma membrane receptor integrin αvβ3 [15,16].

Besides, the metabolism of cancer cells has recently obtained increasing attention due to its emerging role in the occurrence and development of cancers, and endocrine-related tumors are no exception. It refers to glycolysis, lipid metabolism, glutamine metabolism, and so on. The activated aerobic glycolysis (Warburg effect) in the cancer cells is related to oncogenic effectors and loss of tumor suppressors. Inhibition of cancer cell glycolytic capacity has become a promising strategy for anti-tumor research [17]. Cancer cells also utilize lipid metabolism to provide their energy source, membrane components, and signaling molecules for proliferation. Cytoplasmic acetyl-CoA for lipid synthesis is produced from the ATP citrate lyase (ACLY)-catalyzed citrate and acetyl-CoA synthetase (ACSS)-catalyzed acetate, which is supplied by glucose and glutamine in the tricarboxylic acid (TCA) cycle and reductive carboxylation [18]. Interestingly, it has been found that cancer cells can also use glutamine as a carbon source (not emphasized in the Warburg effect) and glutamine metabolism provides intermediates in the TCA cycle to support other biosynthetic pathways [19].

Glutamine (Gln) is one of the dispensable amino acids as it can be synthesized by the human body stably, but some cells are dependent on Gln, and they will die under glutamine depletion, including both normal and cancer cells. It contributes as an energy source for biosynthesis and homeostasis, especially under catabolic stress conditions and is consumed by the kidney, gastrointestinal tract, and immune compartment [20,21,22,23]. As shown in Figure 1, glutamine metabolism can fuel ATP production with participation in the TCA cycle. It enters mammalian cells with the help of a vital amino acid transporter, SLC1A5 (also called ASCT2), and then is transformed into glutamate (Glu) in the mitochondria through a deamination reaction with the catalysis of two different kinds of glutaminase, GLS (kidney type glutaminase) and GLS2 (liver type glutaminase). Glu is then converted to α-ketoglutarate (α-KG), catalyzed by one of the two enzymes: glutamate dehydrogenase (GLUD1 or GLUD2) or aminotransferase. α-KG enters into the TCA cycle and participates in the generation of ATP with production of NADH and FADH_2_. In addition, malate, one of the TCA cycle intermediates, leaves the cycle to produce pyruvate and NADPH. Another product, oxaloacetate (OAA), can be converted to aspartate for nucleotide synthesis. Citrate will further participate in the synthesis of acetyl-CoA and lipids through the cataplerosis of citrate [20,21,22,23,24,25,26,27,28,29]. Glutamine also serves as a biosynthetic precursor and reducing agent in regulating the amino acid pools. In addition to glutamate, it also acts as a nitrogen donor for the synthesis of other dispensable amino acids, including alanine, aspartate, and serine, under the catalysis of aminotransferases like alanine aminotransferase (ALT, also known as GPT, glutamate–pyruvate transaminase), aspartate aminotransferase (AAT, also known as GOT, glutamate–oxaloacetate transaminase), and phosphoserine aminotransferase 1 (PSAT1) [20,21,22,23,24]. Moreover, Gln-derived glutamate can interact with cystine through the homodimer of SLC7A11 and SCL3A2, and glycine is further added to the complex with glutathione synthetase, which finally forms the tripeptide glutathione, showing inhibition of the reactive oxygen species (ROS) by neutralization of peroxide free radicals. Although ROS-mediated cell signaling pathways have shown pro-oncogenic effects physiologically, excess levels of ROS can damage the macromolecules. As mentioned above, glutamine metabolism also produces NADPH, which is necessary for the conversion of oxidized glutathione to reduced glutathione that neutralizes ROS [28,29,30,31]. 

Arguably, glutamine metabolism is indispensable for tumor proliferation, and with the emerging understanding of the role of hormones in Gln metabolism and endocrine oncology, we, therefore, decided to dive deeper into these issues to untangle the connection among the three. Here, we review the specific mechanisms and functions of glutamine metabolism in endocrine-related tumors. Additionally, we also discuss how hormones may contribute to the process and corresponding treatment for endocrine-related cancers.

In mammalian cells, SLC1A5 transports glutamine into the cells while LAT1 mediates the efflux of glutamine from the cells for leucine uptake. This “glutamine recycling” will activate mTOR via RAG and ARF1. Intracellular glutamine can be converted to glutamate by mitochondrial enzyme GLS1 or GLS2, and then glutamate is further catabolized to α-KG, either via GLUD or aminotransferase. α-KG enters into the TCA cycle, during which ATP is produced, and its intermediate, malate, leaves the cycle to generate pyruvate and NADPH. OAA can also be converted to aspartate to join in the generation of pyruvate and NADPH. Citrate further participates in the synthesis of lipids. Meanwhile, glutamate helps for the synthesis of alanine, aspartate, and serine under the catalysis of aminotransferase GPT, GOT, and PSAT1. It is also directly or indirectly involved in the biosynthesis of glutathione, which can be converted from GSSG with NADPH consumed and helps for ROS scavenging. Besides, c-MYC can regulate the activity of SLA1C5, GDH, and aminotransferase to influence glutamine metabolism. Abbreviations: LAT1: L-type amino acid transporter 1; SLC1A5: solute carrier family 1 member 5; SLC7A11: solute carrier family 7 member 11; SCL3A2: solute carrier family 3 member 2; GLUD: glutamate dehydrogenase; GLS: glutaminase; GOT: glutamate–oxaloacetate transaminase; OAA: oxaloacetate; TCA: tricarboxylic acid; NADPH: reduced nicotinamide adenine nucleotide phosphate; FADH2: reduced flavin adenine dinucleotide; ATP: adenosine triphosphate; α-KG: α-ketoglutarate; GPT: glutamate–pyruvate transaminase; PSAT1: phosphoserine aminotransferase 1; ARF1: ADP ribosylation factor 1; RAG: Ras-related GTPase; GSSG: glutathione disulfide; ROS: reactive oxygen species.

## 2. Glutamine Metabolism Characteristics in Endocrine-Related Cancers

Gln is a non-essential amino acid (NEAA) abundant in the human body, and high glutamine consumption has been discovered in various cancers, including pancreatic cancer, acute myeloid leukemia, and endocrine-related cancers (e.g., ovarian cancer, breast cancer, thyroid cancer) [32,33,34,35,36,37,38]. Herein, we try to discuss the glutamine metabolism characteristics in the endocrine-related cancers.

### 2.1. Incorporation into the TCA Cycle

In endocrine-related cancers, cancer cells uptake glutamine via ASCT2 [34,39,40,41]; intracellular glutamine can be utilized first by converting to glutamate catalyzed by mitochondrial enzyme GLS, and then glutamate is further catabolized to TCA cycle substrate α-KG, either via GLUD or transaminase [40,42,43,44,45,46]. As Gln feeds into the TCA cycle, ATP is generated and energy is supplied to cancerous cells. It is unrestrained proliferation, unbridled implantation, and invasive cancer cells’ capabilities that are bound to be accompanied by an increase in ATP and high energy requirements. As expected, Gln has been reported to become the primary carbon source for the maintenance of the TCA cycle in aggressive, therapy-resistant, and advanced prostate cancer [47]. Additionally, evidence has been provided that Gln significantly increases TCA cycle metabolite abundances in highly invasive ovarian cancer cells (SKOV3) compared to lowly invasive ovarian cancer cells (OVCAR3), and Gln addition doubles the ATP content in highly invasive subtypes [33]. Alternatively, the corresponding transporters and enzymes have been found to be highly expressed in these cancer patient samples, such as SLC1A5 [48], GLS [35,38,46], and GLUD1 [49]. Therefore, as with other cancer types, glutaminolysis is also a considerable metabolic pathway of endocrine-related cancer cells that provides TCA cycle intermediates, increases the TCA cycle process, and gives more energy.

### 2.2. Fatty Acid Synthesis

It has been reported that the LNCaP cell, an androgen-sensitive line of prostate cancer cells, can directly utilize glutamine for fatty acid production, and so glutamine was identified as a precursor of fatty acids and cholesterol in PC cells [48], which has been confirmed by subsequent studies [50,51]. This pathway may be due to a process referred to as reductive carboxylation. During the course, a portion of the TCA cycle reverses itself, transforming α-KG into citrate, which can then be metabolized into cytosolic acetyl-CoA, participating in the de novo lipogenesis in the oncogenic contexts with a deficit of glucose-derived acetyl-CoA [52,53,54].

### 2.3. NADPH Synthesis

Yang et al. claimed that Gln acts as an alternative fuel for NADPH, which decreases ROS, in highly invasive OVCA cells (SKOV3 and SKOV3ip), but not in lowly invasive OVCA cells (OVCAR3) [33]. This procedure has also been obtained in mouse breast cancer cells (4T1) [55] and human prostate cancer cells (PC-3) [56]. It has been reported that NADPH is a designated donor of reducing equivalents, generated from NADP+, used in cellular biosynthetic reactions. As it is indispensable for sustaining reductive biosynthesis and maintaining redox homeostasis, a proliferating cell must allocate part of its carbon substrates to be applied to NADPH generation [57,58]. Accumulating data suggest that oxaloacetate from the oxidation of glutamine-derived a-ketoglutarate can be catabolized to malate. Following this, malate can be further oxidized to pyruvate in a reaction that concurrently produces NADPH under glucose-deprived conditions. The synthesis process has been found in tumor cells, including ovarian, breast, and prostate cancers [28,55,59].

### 2.4. Glutathione Synthesis

Recent studies have found that Gln-derived Glu and α-KG are essential for the biosynthesis of glutathione (GSH), ROS scavenging, and protection against oxidative stress in prostate cancer cells, and glutaminolysis thus serves as a critical regulator of prostate cancer radiosensitivity and radioresistance [60,61]. GSH, a prime cellular antioxidant tripeptide, consists of glutamate, cysteine, and glycine. Therefore, in addition to fueling the TCA cycle as described above, Gln-derived glutamate also partially participates in the synthesis of GSH [62,63,64]. In particular, it as been suggested that NADPH can be consumed by tumor cells as the substrate with the conversion of glutathione disulfide (GSSG) to GSH by glutathione reductase (GR) [33,65,66]. Collectively, by Gln breaking down, GSH is generated, accounting for the maintenance of redox status and protecting cells from oxidative stress.

### 2.5. Facilitating Leucine Transport

SLC1A5, mentioned above, functions as an antiporter that transports Gln into the intracellular area, whereas SLC7A5 (LAT1) then mediates the efflux of Gln from the cells to facilitate uptake of its substrate leucine, an essential amino acid [67,68]. Given that intracellular leucine and SLC7A5 have been proven to be responsible for the growth of ER+ breast cancer cells [69], intracellular glutamine is aberrantly exported from the cell to support the leucine-dependent cell proliferation [40]. Besides, this “glutamine recycling” has been found in PC-3 cells as well. The researchers’ explanation for this is the activation of the mammalian target of rapamycin complex 1 (mTORC1), which is stimulated under the activation of the Ras-related GTPase (RAG) complex and recruited on lysosomes facilitated by ADP ribosylation factor 1 (ARF1) and may indirectly affect mitochondrial respiration and favor cell growth [48].

## 3. Hormone Regulation of Glutamine Metabolism

Hormones have shown different roles in participating in the glutamine metabolism and, therefore, in regulating further biological responses. Here, we will discuss the effects of hormone regulation on glutamine metabolism, including estrogen, progesterone, androgen, prostaglandin, and thyroid hormone, as shown in Table 1.

### 3.1. Estrogen and Progesterone

Glutamate and glutamine remain balanced in the human body under the synthesis of glutamine catalyzed by glutamine synthetase and hydrolysis of glutamine with GLS catalysis. The conversion of glutamate to glutamine mainly happens when glutamate is uptaken by astroglial cells, which are located in glial cells in neural system under the catalysis of glutamine synthetase. The glutamine formed in astroglia is transported with facilitated diffusion via Na^+^ and H^+^-coupled electroneutral systems—N transporters. Research by Haghighat et al. has proven that estrogen enhances glutamine synthetase activity and glutamine formation in C-6 glial cells, which furthermore influences glutamine metabolism. Vallejo et al. suggest that progesterone displays neuroprotective effects and the glutamine synthetase is significantly increased with the treatment of progesterone. Grasso et al. also showed that progesterone interacts with glucocorticoid receptors and induces glutamine synthetase [70,71,83,84,100]. Studies from Zlotnik et al. proved that the blood glutamate level is inversely correlated to the increasing level of plasma estrogen and progesterone during pregnancy and menstrual cycles [101]. Interestingly, there has also been research that found that mitochondrial glutamine and glutamate metabolism may contribute to progesterone biosynthesis by providing NADPH [72].

Estrogen can also directly facilitate glutamine metabolism by increasing aminotransferase activity. Research by Zhou et al. proved that estrogen can increase Gln metabolism of estrogen-sensitive uterine endometrial carcinoma cells by upregulating the c-MYC pathway and, therefore, increasing the GLS level, which transfers cellular glutamine to glutamate [46]. The oncogene, c-MYC, as a master transcriptional factor, modulates approximately 10–15% of genes in the genome and regulates the expression of SLC7A11, SLC1A5, SLC6A14, SLC7A5, GLS, and so on [73,74,75]. A study carried out by Chen et al. also found that for ER-positive and aromatase inhibitor-sensitive breast cancers, estrogen can bind to ER and further upregulate the c-MYC pathway to increase glutamine metabolism. Interestingly, although glutamine metabolism is independent of estrogen in aromatase inhibitor-resistant breast cancer, the cross-talk between ER and HER2 upregulates the c-MYC pathway and further increases the expression level of GLS, SLC1A5, and glutamine consumption. In line with Chen’s study, estrogen has been proven to promote ovarian cancer development through either receptor-dependent or receptor-independent pathways by binding to the estrogen receptor α and activating estrogen transcriptive genes like c-MYC, HER2, and growth factors, which lead to cell differentiation and division [76,77,78,79]. Estrogen-related receptors (ERR) can also control glutathione production and downstream regulate glutamine metabolism participation in the TCA cycle and also influence ROS, which is essential in cancer development [80,81,82,102]. On the other hand, a weak positive correlation between GLS with PI3KCa within the low proliferative luminal breast cancer was put forward by Masisi’s group. PIK3Ca mutations, shown to be linked to hormone receptor-positive breast cancer [103], are involved in metabolic reprogramming, specifically enhancing glutamine uptake and glutamate generation via regulation of pyruvate dehydrogenase activity [104]. Therefore, glutamine metabolism can be regulated in estrogen-dependent manners. Additionally, the synergistic and/or antagonistic effects and detailed mechanisms of progesterone and estrogen in glutamine metabolism need to be further researched.

### 3.2. Androgen and Prostaglandin

It has been suggested that androgen and prostaglandin are implicated in carcinogenesis and the progression of a lot of cancers, especially prostate cancer, and almost all prostate cancers are androgen dependent [90,105]. Barfeld et al. have reported that androgen regulates biosynthesis and metabolism in PC, including the c-MYC signaling pathway [93]. As androgen is stimulated, the expression of c-MYC in PC cells will be suppressed [91]. The steroid-activated androgen receptor (AR) can also decrease the c-MYC signaling pathway, which accounts for driving reinforced glutaminolysis in cancer cells through regulating glutamine transporters and GLS1 directly or indirectly (via miRNA) [74,92,93]. However, White’s team claimed that AR signaling could upregulate the expression of glutamine transporters SLC1A4 and SLC1A5, which are overexpressed in prostate cancer. MYC only acts as a context-dependent regulator of SLC1A4 and SLC1A5 levels while mTORC1 is involved in the maximum AR-mediated glutamine uptake [106]. Of note, Xu and his colleague found that AR upregulated GLS1 and increased the glutamine utilization, and therefore, androgen deprivation is considered as an important method to prevent glutamine uptake by cancer cells [47]. More interestingly, the gut microbe has been reported to be involved in androgen-regulated glucose homeostasis and circulating glutamine/glutamate ratio [94].

In addition, prostaglandin E2 (PGE2) can mediate the Ca2+-dependent release of glutamate from neuron cells under the coactivation of the AMPA/kainate and metabotropic glutamate receptors (mGluRs) [95,96]. GABA-transaminase (GABA-T) can convert glutamate (Glu)–γ-aminobutyric acid (GABA) to succinate as a TCA cycle intermediate and further convert α-KG to resynthesize intramitochondrial glutamate for the TCA cycle, and AAT converts glutamate to α-KG and aspartate. Notably, prostaglandins E1 (PGE1) can induce morphological changes in astrocytes with dibutyryl cyclic AMP (dBc AMP) and further increase two enzymes, GABA-T and AAT, in the GABA cycle and glutamate metabolism [97,98,107].

### 3.3. Thyroid Hormones

Thyroid hormone signaling is necessary for skeletal muscle development and research has proven that alanine and glutamine are synthesized and released in skeletal muscle, indicating that the thyroid hormone may contribute to the regulation of glutamine metabolism [85,108]. Glutamine synthetase is prominent in converting glutamate to glutamine, as discussed before, and it is developmentally regulated in the oligodendrocyte lineage. T3 has been found to control glutamine synthetase levels and, therefore, regulate oligodendrocyte maturation and progenitor cell proliferation [86]. Cicatiello et al. also suggested that the thyroid hormone regulated glutamine metabolism by upregulating the GPT2 gene. GPT2 is an aminotransferase that transfers nitrogen from glutamate to pyruvate to make alanine and α-ketoglutarate. In turn, the mutation of glutamine also interferes with the function of thyroid hormone nuclear receptors [87,109]. Besides, research by Ou et al. described a positive association between GABA and T3, as well as Gln and T4 concentrations in the serum of manganese-exposed rats [110]. Parry-Billings et al. held the hypothesis that the thyroid hormone can influence glutamine release by skeletal muscle. More specifically, T3 induces the rate of glutamine release from skeletal muscle, through which, T3 may affect the function of the immune system [111,112].

Other thyroid-related hormones also influence glutamine metabolism. Thyroid-stimulating hormones are produced by pars tuberalis (PT). Aizawa et al. found that the ionotropic glutamic acid receptor (iGluR) KA2 was highly expressed in PT, as well as glutamine transporter (ATA2) and GLS2. These findings suggest that glutamine is highly ingested by PT and converted to glutamate by GLS [88]. Garber et al. showed that a high level of parathyroid hormones has direct effects on skeletal muscle formation. The parathyroid hormone regulates glutamine metabolism by increasing glutamine uptake and using GLS1 to convert glutamine to glutamate. The parathyroid hormone can also augment glutamine utilization through both phosphate-dependent glutaminase (PDG) and GDH pathways to increase α-KG formation in glutamine metabolism and increase extracellular glutamate transport [89,99].

### 3.4. Insulin

Insulin has been found to boost the expression of glutamine synthetase (GS) mediated by sterol regulatory element-binding protein 1 (SREBP1), and increased GS expression further enhances glutamine-dependent anabolic pathways to promote nucleotide and protein synthesis, lipid droplet (LD) formation, and lipogenesis in breast cancer cells [113]. Therefore, insulin can impact on breast cancer cell growth, proliferation, and migration, which has also been proved by other studies [114,115].

## 4. Functional Roles of Glutamine Metabolism on Endocrine-Related Cancers

With the field of glutamine metabolism becoming a hotspot for research, it has become evident that Gln metabolism plays a prominent role in energy production, biosynthesis, and redox homeostasis, and so is vital for establishing and maintaining a tumorigenic status by becoming involving in cancer-related signaling pathways, such as c-MYC, mTOR, Akt, Ras, and AMPK pathways [23,63,116,117,118,119,120,121]. Then, what about endocrine tumors? We will further clarify the specific roles of glutamine metabolism in endocrine-related cancers as follows and as shown in Figure 2 and Table 2.

(1)Gln metabolism is able to protect endocrine-related cancer cells against apoptosis and autophagy through the TIGAR and STAT3 signaling pathways [33,122].(2)Gln metabolism helps tumor cell growth and proliferation via the c-MYC and mTORC1 signaling pathways, and by producing ATP and fatty acid [92,123,124,125].(3)Gln metabolism can generate NADPH and GSH to decrease the generation of ROS and maintain the redox state [62,63,126,127].(4)Gln metabolism can act on immature myeloid cells, CD11b+Gr1+ myeloid cells, macrophages, and T lymphocytes to realize antitumor immune response [128,129,130,131,132].

Abbreviations: Gln: glutamine; Leu: leucine; ASCT2: amino acid transporter-2; LAT1: L-type amino acid transporter 1; TIGAR: TP53-induced glycolysis and apoptosis regulator; STAT3: signal transducer and activator of transcription 3; mTORC1: mammalian target of rapamycin complex 1; GLS: glutaminase; TCA: tricarboxylic acid; ATP: adenosine triphosphate; NADPH: reduced nicotinamide adenine nucleotide phosphate; GSH: glutathione; ROS: reactive oxygen species; NAA: N-acetylaspartate.

### 4.1. Breast Cancer

Glutamine metabolism has been shown to favor proliferation and be associated with aggressive breast cancer [35,133,134,135], which is partly due to the hyperactivation of mTORC1, a critical regulator of cell signaling and metabolic pathways, promoting protein translation and fatty acids biosynthesis, improving cell growth, and inhibiting catabolism or autophagy [117,136]. Another important reason may be that glutaminolysis supplies the cancer cells with intermediates for the synthesis of macromolecules, redox balance, and mitochondrial energy metabolism required for growth [34,39,137,138,153]. In addition, it has been demonstrated that glutamine can augment mitochondrial mass and decrease the expression of autophagy markers in the human breast adenocarcinoma cell line (MCF7 cells). Moreover, Gln is able to protect MCF7 cells against apoptosis via the upregulation of TIGAR, a multi-functional protein that suppresses glycolysis, apoptosis, and autophagy [122]. Of note, Wu et al. have suggested that the metabolism of glutamine is the master metabolic pathway controlling the proliferation and activation of immature myeloid cells using a patient-derived and 4T1 breast cancer preclinical model [128]. It has also been found that restraint of glutaminolysis in 4T1 breast cancer preclinical model contributes to immunogenic tumor cell death [139]. In triple-negative breast cancer (TNBC), glutamine metabolism has been discovered to refrain tumor-infiltrating T lymphocyte numbers and function, and so the team concluded that increased glutamine metabolism combined with the declined cytotoxic potential of T cells within tumors may result in the nonresponsiveness of a significant proportion of TNBC patients to immunotherapies [129,130]. Besides, Dias et al. also showed that in TNBC, the GLS2 level increased significantly and sequentially promoted tumor proliferation and metastasis. These reports suggest that an active glutamine metabolism promotes the development of TNBC under the regulation of non-hormone factors. Gln metabolism is also enhanced to facilitate the growth of advanced hormone receptor-positive breast cancer, the development of which is independent of estrogen [35,140,141]. By considering the above-mentioned facts, it is clear that Gln metabolism is conducive to the propagation, anti-apoptotic and anti-autophagic ability, and immune resistance of breast cancer, thus hastening tumor progression.

### 4.2. Uterine Endometrial Carcinoma

Nowadays, the emerging understanding of estrogen and glutamine in regulating the development of uterine endometrial carcinoma (UEC) has been established [46,142,143]. The intracellular glutamine has been described as mediating the ability of LAT1 to enhance the growth of UEC cells. LAT1, a reciprocal transporter, plays a growth-promoting role in UEC [154] via uptake of leucine in exchange for efflux of glutamine, which further activates the mTORC1 pathway to facilitate the proliferation [123]. Zhou et al. have proposed that estrogen enables the activation of glutaminolysis, which in turn suppresses autophagy in endometrial cancer cells of estrogen-sensitive UEC cell line Ishikawa cells [46]. This process relies on the upregulation of GLS, activated by c-MYC under the regulation of estrogen, which is consistent with previous reports [92]. Additionally, they found Gln can promote the growth of both Ishikawa and KLE cells regardless of estrogen sensitivity. In Ishikawa cells, estrogen’s ability to improve cellular viability is an accounting factor for this effect [155]. Overall, glutamine metabolism accelerates tumor growth and inhibits autophagy in uterine endometrial cancer.

### 4.3. Ovarian Cancer

Ovarian cancer (OVCA) is regarded as the highest mortality of gynecological malignancies, for the reason that most metastases have already occurred with a poor prognosis by the time they are detected [144,145,156], for which glutamine metabolism occupies a nonnegligible position. First of all, Gln metabolism has been extensively elucidated to not only help in the proliferation and metastasis of OVCA cells, but also has been implicated in ovarian cancer aggressiveness and increased invasion [20,33,54,124,146,147,148]. In addition to promoting biosynthesis, energy supply, relieving oxidative stress, and maintaining redox, the mTOR/S6 and MAPK pathways also play an important role, as inhibiting mTOR activity by rapamycin or blockade of S6 expression via siRNA will restrict the activity of GDH and GLS, resulting in a reduction in Gln-induced cell proliferation [124,125]. Of note, Gln is capable of pushing OVCA cells from stage G1 to stage S to affect cell cycle progression and ultimately induce cell growth [124]. Besides, the activation of STAT3 by glutamine in OVCA has been already unraveled, and it has been reported that Gln deprivation regulates STAT3 phosphorylation, thereby causing the cancer metabolic switch [33]. STAT3 has been shown to function in cell differentiation, antiapoptotic response, cancer metastasis, and regulating cancer hallmarks, such that it potentiates drug resistance and correlates with highly invasive OVCA cells [147,149,150,151,152,157,158]. Recently, Menga’s team uncovered the unprecedented role of glutaminolysis in regulating macrophage polarization in highly invasive OVCA and they clarified that this effect links to N-acetylaspartate (NAA) synthesis as its anti-inflammatory and protumoral properties [131]. Furthermore, glutamine metabolism has also been proven to modulate the metabolism and function of immunosuppressive CD11b+Gr1+ myeloid cells in ovarian cancer, finally influencing the immune environment of tumors and the antitumor immune response [132]. According to the above analysis, we suppose that the proliferation, migratory and invasive abilities, immune resistance, and anti-apoptosis of OVCA cells are dependent on glutaminolysis to some extent.

### 4.4. Prostate Cancer

Prostate cancer (PC) is a hormone-dependent disease, known for its metabolism deregulation, metabolic rewiring, and reliance on glutamine [159,160,161,162,163]. Glutamine has been previously found to decrease the generation of ROS and produce GSH in PC to maintain the redox state, reduce the risk of oxidative stress, and, more importantly, mediate PC radioresistance [60,61,126,127]. In addition, the biosynthesis and energy production of glutamine metabolism can be observed in PC, and consequently, glutaminolysis has been found to be involved in the cell growth, division and invasion, and the progression of the tumor [47,48,50,51,60]. To conclude, glutamine metabolism benefits prostate cancer mainly in promoting the development of the tumor and radiation resistance.

### 4.5. Thyroid Cancer

In thyroid cancer (TC), the alteration of glutamine level and Gln metabolism-related protein expression have been demonstrated to be implicated in tumorigenesis [38,164]. Like in other endocrine related cancers, glutaminolysis plays a prominent role in the development and progression of papillary thyroid cancer (PTC), owing a great deal to the key enzyme, GLS [38]. It has been well described that blocking GLS can inhibit glutamine metabolism; limit mitochondrial respiration; induce apoptosis and autophagy via the mTORC1 signaling pathway; and end with the growth, viability, and invasion of PTC cells getting impaired. Accordingly, glutaminolysis is associated with the formation and progression of TC. However, given the limited number of studies to date, further research is needed and the relevant mechanism remains to be elucidated.

## 5. Hormone/Glutamine Metabolism Axis in Endocrine-Related Cancer Intervention

Currently, the use of hormone/glutamine metabolic axis for anti-tumor therapy has become increasingly common, as listed in Table 3. CB-839, a GLS inhibitor, is regarded as a putative treatment for tamoxifen-resistant LCC9 breast cancer cells [35] and UECC [46]. It has been found that CB-839 effectively increases glutamine in LCC9 cells, which have been proved to be dependent on upregulated pro-survival autophagy [165], and so it is estimated that the efficacy of CB-839 links to the disruption of amino acid metabolism following autophagy [35]. As for the uterine endometrial carcinoma cell (UECC), CB-839 significantly suppresses glutaminolysis and abrogates estrogen-induced cell viability and the inhibitory effect of autophagy in estrogen-sensitive UECC in vitro and in vivo [46]. Alternatively, chemical inhibition or shRNA knockdown of ASCT2 has been shown to be an effective therapeutic target in PC. Given that ASCT2 is androgen-regulated, this intervention will target PC at multiple sites along the androgen/glutamine metabolism axis [48]. Meanwhile, the glutaminase inhibitor BPTES can decrease the viability and migration of PC cells while increasing caspase-3 activity. There is an additive effect on suppressing androgen-sensitive LNCaP cell viability when BPTES and anti-androgen bicalutamide coexist [166]. Besides, chemical inhibition of glutamine transport by benzylserine (BenSer) or GPNA has also been proved to work by limiting UECC growth [142]. BenSer has also been described as inhibiting breast cancer cell growth and viability by blocking ASCT2 and LAT1 activity [167]. To sum up, Gln metabolism as a target of intervention in the treatment of endocrine-related cancers has potential value and deserves further exploration.

## 6. Conclusions

In conclusion, as shown above, hormone-regulated glutaminolysis is involved in various aspects of endocrine-related cancers, such as tumor growth, invasion, oxidative stress, autophagy, and so on. The hormone/glutamine metabolism axis plays a momentous role in tumor genesis and development, and the targeted intervention measures have already achieved certain progress in tumor treatment. Through the summarized literature review of these findings, we believe that future research directions in this field can focus on the aspects as follows. Firstly, although there have been some studies on the regulatory effect of glutamine metabolism on endocrine oncology, the local regulatory mechanism is still inadequate and needs further study. Secondly, the regulation of glutamine metabolism on the immune microenvironment of endocrine related cancers and related mechanisms are also worth exploring. Thirdly, there are many other factors (e.g., hypoxia, epigenetics, cytokines) involved in the regulation of glutamine metabolism to affect non-endocrine-related cancers and endocrine-related cancers. Hormones can be regarded as a therapeutic target due to their regulatory effects on glutamine metabolism in endocrine-related cancers, which is different to non-endocrine-related cancers. Therefore, more drug development efforts ought to be made to exploit the novel therapeutic strategies targeting glutamine metabolism, especially in combination with hormone-targeted interventions, for patients with endocrine-related cancer.

## Figures and Tables

**Figure 1 ijms-23-10086-f001:**
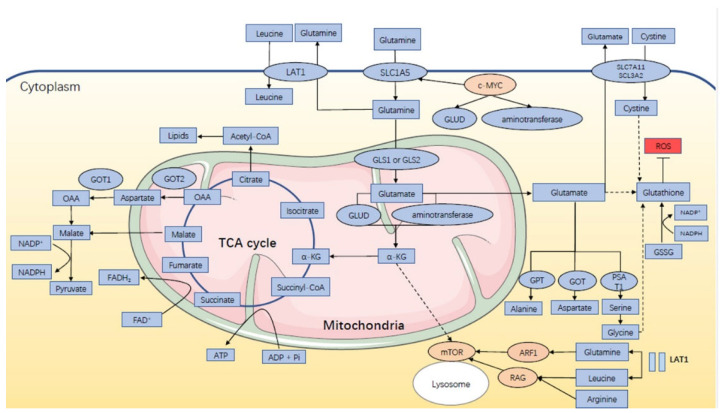
Glutamine metabolism in endocrine-related cancers.

**Figure 2 ijms-23-10086-f002:**
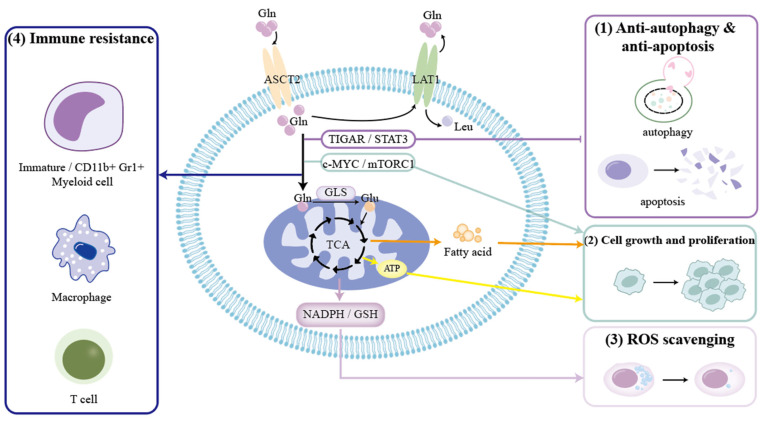
Effect of glutamine metabolism on endocrine-related cancers.

**Table 1 ijms-23-10086-t001:** Effects of regulation of hormones on glutamine metabolism.

Hormones	Enzyme or Gene	Influence	References
Estrogen	Glutamine synthetase	Estrogen enhances glutamine synthetase and increases glutamine formation in glial cells in the neural system	[69,70,71]
	GLS	Estrogen increases GLS levels in breast cancer	[72,73,74]
	GLUD	Estrogen increases GLUD levels in breast cancer	[72,73,74]
	c-MYC	Estrogen upregulates the c-MYC pathway in endometrial cancer and breast cancer	[46,72,73,74,75,76,77,78]
	SLC1A5	Estrogen increases SLC1A5 gene expression in breast cancer	[72,73,74]
	Glutathione	Estrogen receptors control glutathione production and downstream regulate glutamine metabolism in the TCA cycle and ROS in breast cancer	[79,80,81,82]
Progesterone	Glutamine synthetase	Progesterone interacts with glucocorticoid receptors and induces glutamine synthetase	[83,84]
Thyroid hormone	Glutamine synthetase	Thyroid hormone controls glutamine synthetase levels in oligodendrocyte lineage	[85]
	GPT2	Thyroid hormone upregulates mitochondrial GPT2 in skeletal muscle	[86,87]
Parathyroid Hormone	GLS1	PH upregulates GLS1 in skeletal muscles	[88]
	GLDH (GLUD)	PH stimulates the GLDH pathway in the renal system	[89]
	PDG	PH increases glutamine utilization through the PDG pathway in renal system	[89]
Androgen	GLS1	AR upregulates GLS1 in prostate cancer	[47,73,90]
	SLC1A4, SLC1A5	Androgen receptor signaling upregulates glutamine transporters in prostate cancer	[91]
	c-MYC	Androgen receptors decrease the c-MYC signaling pathway, which regulates glutamine transporters and GLS1 in prostate cancer	[92]
	mTORC1	mTORC1 is needed for maximum AR-mediated glutamine uptake in prostate cancer	[90,93]
Prostaglandin (PGE2)	AMPA/kaniate and mGluRs	PGE2 coactivates AMPA/kainate and mGluRs to mediate Ca^2+^ dependent glutamate release from neuron cells	[92]
Prostaglandin (PGE1)	GABA-T	PGE1 induces morphological changes in astrocytes and increases GABA-T	[94,95]
	AAT (or GOT)	PGE1 induces morphological changes in astrocytes and increases AAT	[96,97,98]
**Insulin**	Glutamine synthetase	Insulin increases the expression of GS in breast cancer	[99]

Note: abbreviations: GLS: glutaminase; GLUD: glutamate dehydrogenase; SLC1A5: neutral amino acid transporter B(0) (also known as ASCT2); GPT2: glutamate–pyruvate transaminase (mitochondrial isoform, also known as alanine aminotransferase); GLS1: kidney-type glutaminase; GLDH (GLUD): glutamate dehydrogenase; PDG: phosphate-dependent glutaminase; SLC1A4: neutral amino acid transporter A; AMPA: α-amino-3-hydroxy-5-methyl-4-isoxazolepropionic acid receptor; mGluRs: metabotropic glutamate receptors; GABA-T: GABA-transaminase; AAT (or GOT): aspartate aminotransferase (also known as glutamate–oxaloacetate transaminase, GOT).

**Table 2 ijms-23-10086-t002:** Influence of glutamine metabolism on endocrine-related cancers.

Endocrine-Related Cancers	Proliferation and Invasion	Autophagy and Apoptosis	Oxidative Stress	Other Functions	Tumor Outcome	References
Breast cancer	Promotes proliferation and invasion	Inhibits autophagy and apoptosis	Maintains redox balance	Immune resistance	Promotes progression	[34,35,39]
Uterine endometrial carcinoma	Accelerates tumor growth	Inhibits autophagy	Not mentioned	Not mentioned	Promotes progression	[121,127,128,131,132,133,134,135,136,137,138,139]
Ovarian cancer	Promotes proliferation and invasion	Inhibits apoptosis	Relieves oxidative stress	Immune resistance, drug resistance	Promotes progression	[46,91,122,140,141,142,143]
Prostate cancer	Promotes proliferation and invasion	Not mentioned	Decreases ROS, produce GSH	Radiation resistance	Promotes progression	[20,33,54]
Thyroid cancer	Promotes proliferation and invasion	Not mentioned	Not mentioned	Not mentioned	Promotes progression	[123,124,129,130,144,145,146,147,148,149,150,151,152]

Note: abbreviations: ROS: reactive oxygen species; GSH: glutathione.

**Table 3 ijms-23-10086-t003:** Drug intervention in endocrine-related cancers.

Intervention	Target	Mechanism	Endocrine-Related Cancers	References
CB-839	GLS	Upregulates autophagy, abrogates estrogen-induced cell viability	Tamoxifen-resistant LCC9 BC, UEC	[35,46,163]
Chemical inhibition/shRNA knockdown of ASCT2	ASCT2	Inhibits the androgen/glutamine metabolism axis	PC	[48]
BPTES	GLS	Decreases the viability and migration of cell, increases caspase-3 activity	PC	[164]
BenSer	ASCT2/LAT1	Inhibits cell growth and viability	UEC, BC	[140,165]
GPNA	ASCT2	Inhibits cell growth	UEC	[140]

Note: abbreviations: GLS: glutaminase; BC: breast cancer; UEC: uterine endometrial carcinoma; ASCT2: amino acid transporter-2; PC: prostate cancer; LAT1: L-type amino acid transporter 1.

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
