# Peer review of "Hormone-Glutamine Metabolism: A Critical Regulatory Axis in Endocrine-Related Cancers"

_ijms, 2022, doi:10.3390/ijms231710086_

Round 1
Reviewer 1 Report
This review discusses the critical roles of hormone and glutamine metabolism in endocrine-related cancers. Specific comments are as follows:
1. Abbreviations used in the text should be defined in the text at first use, such as ACYL and ACSS on page 2.
2. “SLA1C5” in Fig. 1 should be changed to “SLCIA5”.
3. In Fig. 1, the metabolic pathways and metabolites should be drawn within or out of mitochondria clearly.
4. On page 3, section 2.1, glutamate dehydrogenase “(GDH)” should be changed to GLUD since glutamate dehydrogenase is mentioned as GLUD in the introduction.
5. Endocrine-related cancers generally indicate sex steroid-responsive cancers, including breast cancers, endometrial cancers, prostate cancers, testis cancers, thyroid cancers, and ovary cancers. However, the term “endocrine-related cancer” has been defined in a more expanded way, in which “endocrine-related cancer” are also regulated by some metabolic hormones, such as insulin (Belfiore and Perks, Front Endocrinol (Lausanne) 2013, 4:141). Insulin can promote breast cancer cell proliferation, survival, and migration (Biello et al., Biomolecules 2021, 11(1), 125). Moreover, insulin is shown to increase the expression of glutamine synthetase and enhanced glutamine synthetase expression boosts glutamine-dependent anabolic pathways to promote lipogenesis in breast cancer cells (Jhu et al., International Journal of Molecular Sciences 2021, 22(18), 9814). The authors are suggested to include insulin-induced glutamine synthetase in breast cancer cells in table 1.
6. In line 423, please define the term “UECC”. Besides, “UEC” is shown in table 3. Please keep the consistency.
7. In table 1, 2 and 3, references are suggested to be cited and shown in the column directly for convenient searching. In the figure legend of Fig. 2, references are suggested to be cited right after the description (1)-(4).
Author Response
Dear editors,
Thank you very much for your letter enclosing the comments for our manuscript entitled “Hormone-Glutamine Metabolism: A Critical Regulatory Axis in Endocrine-Related Cancers” (1887691). We resubmit a revised manuscript in which the revisions have been labeled in red. The following are the responses to comments and suggestions about the manuscript.
We wish to take this opportunity to express our gratitude for your reconsideration of our paper for publication in your journal, International Journal of Molecular Sciences.
To Reviewer #1:
Comments to the Author
- Abbreviations used in the text should be defined in the text at first use, such as ACYL and ACSS on page 2
Response: The full name for ACLY is ATP citrate lyase and the full name for ACSS is acetyl-CoA synthetase. We have added these to the text.
- “SLA1C5” in Fig. 1 should be changed to “SLCIA5”.
Response: We are grateful for such a careful reading. We now have corrected the related word in Fig. 1.
- In Fig. 1, the metabolic pathways and metabolites should be drawn within or out of mitochondria clearly.
Response: Thank you. We have further revised Fig.1 according to your suggestion as follows. Hope it can meet your requirements.
- On page 3, section 2.1, glutamate dehydrogenase “(GDH)” should be changed to GLUD since glutamate dehydrogenase is mentioned as GLUD in the introduction.
Response: Thank you for your suggestion. We have made two changes in section 2.1 as follows.
Line 134: either via GLUD or transaminase [40, 42-46].
Line 145: and GLUD1 [49].
- Endocrine-related cancers generally indicate sex steroid-responsive cancers, including breast cancers, endometrial cancers, prostate cancers, testis cancers, thyroid cancers, and ovary cancers. However, the term “endocrine-related cancer” has been defined in a more expanded way, in which “endocrine-related cancer” are also regulated by some metabolic hormones, such as insulin (Belfiore and Perks, Front Endocrinol (Lausanne) 2013, 4:141). Insulin can promote breast cancer cell proliferation, survival, and migration(Biello et al., Biomolecules 2021, 11(1), 125). Moreover, insulin is shown to increase the expression of glutamine synthetase and enhanced glutamine synthetase expression boosts glutamine-dependent anabolic pathways to promote lipogenesis in breast cancer cells (Jhu et al., International Journal of Molecular Sciences 2021, 22(18), 9814). The authors are suggested to include insulin-induced glutamine synthetase in breast cancer cells in table 1.
Response: Thank you for your suggestions. We have supplemented the corresponding insulin section as follows.
Page 8, Paragraph 3:
3.4. Insulin
Insulin is found to boost the expression of glutamine synthetase (GS) mediated by sterol regulatory element-binding protein 1 (SREBP1), and increased GS expression further enhances glutamine-dependent anabolic pathways to promote nucleotide and protein synthesis, lipid droplet (LD) formation and lipogenesis in breast cancer cells [114]. Therefore, insulin can impact on breast cancer cell growth, proliferation and migration, which is also proved by other studies [115,116].
Table 1. Regulation of Hormones on Glutamine Metabolism
|
Hormones |
Enzyme or Gene |
Influence |
Reference |
|
Estrogen |
Glutamine synthetase |
Estrogen enhances glutamine synthetase and increase glutamine formation in glial cells in neural system |
71-73 |
|
|
GLS |
Estrogen increases GLS level in breast cancer |
78-80 |
|
|
GLUD |
Estrogen increases GLUD level in breast cancer |
78-80 |
|
|
c-MYC |
Estrogen upregulates c-MYC pathway in endometrial cancer and breast cancer |
46,78-84 |
|
|
SLC1A5 |
Estrogen increases SLC1A5 gene expression in breast cancer |
78-80 |
|
|
Glutathione |
Estrogen receptors control glutathione production and downstream regulate glutamine metabolism in TCA cycle and ROS in breast cancer |
85-88 |
|
Progesterone |
Glutamine synthetase |
Progesterone interacts with glucocorticoid receptor and induce glutamine synthetase |
74, 75 |
|
Thyroid hormone |
Glutamine synthetase |
Thyroid hormone controls glutamine synthetase level in oligodendrocyte lineage |
105 |
|
|
GPT2 |
Thyroid hormone upregulates mitochondrial GPT2 in skeletal muscle |
106,107 |
|
Parathyroid Hormone |
GLS1 |
PH upregulates GLS1 in skeletal muscles |
112 |
|
|
GLDH (GLUD) |
PH stimulates GLDH pathway in renal system |
113 |
|
|
PDG |
PH increases glutamine utilization through PDG pathway in renal system |
113 |
|
Androgen |
GLS1 |
AR upregulates GLS1 in prostate cancer |
47,79,93 |
|
|
SLC1A4, SLC1A5 |
Androgen receptor signaling upregulates glutamine transporters in prostate cancer |
95 |
|
|
c-MYC |
Androgen receptors decrease c-MYC signaling pathway which regulates glutamine transporters and GLS1 in prostate cancer |
96 |
|
|
mTORC1 |
mTORC1 is needed for maximum AR-mediated glutamine uptake in prostate cancer |
93,94 |
|
Prostaglandin (PGE2) |
AMPA/kaniate and mGluRs |
PGE2 coactivates AMPA/kaniate and mGluRs to mediate Ca2+ dependent glutamate release from neuron cells |
96 |
|
Prostaglandin (PGE1) |
GABA-T |
PGE1 induces morphological changes in astrocytes and increase GABA-T |
98,99 |
|
|
AAT (or GOT) |
PGE1 induces morphological changes in astrocytes and increase AAT |
100-102 |
|
Insulin |
Glutamine synthetase |
Insulin increases the expression of GS in breast cancer |
114 |
- In line 423, please define the term “UECC”. Besides, “UEC” is shown in table 3. Please keep the consistency.
Response: Thank you. The full name for UECC is uterine endometrial carcinoma cell and while UEC refers to uterine endometrial carcinoma. We have added it to the text.
- In table 1, 2 and 3, references are suggested to be cited and shown in the column directly for convenient searching. In the figure legend of Fig. 2, references are suggested to be cited right after the description (1)-(4).
Response: Thank you for your comments. We have further refined the tables and the figure legend of Fig. 2.
Reviewer 2 Report
This timely and interesting review by Feng-Yuan Xu, et al. summarized glutamine metabolism characteristics and its influence on endocrine oncology, as well as its upstream hormonal regulation. The authors also discussed the possible therapeutic interventions against hormone-glutamine axis. Overall, this review is well written and comprehensive. It is suitable for publications pending clarification and further discussion of the following points.
1. In section 2, the authors discussed the glutamine metabolism characteristics in the endocrine related cancers, including incorporation into TCA cycle, fatty acid synthesis, NADPH synthesis, glutathione synthesis and leucine transport. However, these characteristics seem to be general pathways for glutamine metabolism. What are the differences between non-endocrine related cancers and endocrine related cancers, in terms of glutamine metabolism?
2. Please add references to each discovery accordingly in Table 1, Table 2 and Table 3.
3. The discussion of hormone-glutamine axis is not in depth. The concept of this review is to conclude how hormone influences glutamine metabolism and then affects tumor growth. However, for example, in section 4.1 breast cancer, in the Triple-negative breast cancers, glutamine itself plays a role in regulating tumor growth. TNBC is not responsive to hormone due to the lack of receptors. Thus, the hormone-glutamine axis does not exist in this type of TNBC. Please modify discussions like that.
Author Response
Dear editors,
Thank you very much for your letter enclosing the comments for our manuscript entitled “Hormone-Glutamine Metabolism: A Critical Regulatory Axis in Endocrine-Related Cancers” (1887691). We resubmit a revised manuscript in which the revisions have been labeled in red. The following are the responses to comments and suggestions about the manuscript.
We wish to take this opportunity to express our gratitude for your reconsideration of our paper for publication in your journal, International Journal of Molecular Sciences.
To Reviewer #2:
Comments to the Author
- In section 2, the authors discussed the glutamine metabolism characteristics in the endocrine related cancers, including incorporation into TCA cycle, fatty acid synthesis, NADPH synthesis, glutathione synthesis and leucine transport. However, these characteristics seem to be general pathways for glutamine metabolism. What are the differences between non-endocrine related cancers and endocrine related cancers, in terms of glutamine metabolism?
Response: Thank you. To our knowledge, in addition to hormones, glutamine metabolism is regulated by many other factors in non-endocrine related cancers, such as hypoxia (Yoo HC, Yu YC, Sung Y, Han JM. Glutamine reliance in cell metabolism. Exp Mol Med. 2020;52(9):1496-1516. doi:10.1038/s12276-020-00504-8), epigenetics (Han H, Feng F, Li H. Research advances on epigenetics and cancer metabolism. Zhejiang Da Xue Xue Bao Yi Xue Ban. 2021;50(1):1-16. doi:10.3724/zdxbyxb-2021-0053), cytokines (Abumrad NN, Kim S, Molina PE. Regulation of gut glutamine metabolism: role of hormones and cytokines. Proc Nutr Soc. 1995;54(2):525-533. doi:10.1079/pns19950021) and so on. Besides, hormones can be regarded as a therapeutic target due to their regulatory effects on glutamine metabolism in endocrine related cancers, which is different to non-endocrine related cancers. We have supplemented these contents in the last section of this manuscript.
- Please add references to each discovery accordingly in Table 1, Table 2 and Table 3.
Response: Thank you for your suggestions. We have further refined the tables.
- The discussion of hormone-glutamine axis is not in depth. The concept of this review is to conclude how hormone influences glutamine metabolism and then affects tumor growth. However, for example, in section 4.1 breast cancer, in the Triple-negative breast cancers, glutamine itself plays a role in regulating tumor growth. TNBC is not responsive to hormone due to the lack of receptors. Thus, the hormone-glutamine axis does not exist in this type of TNBC. Please modify discussions like that.
Response: Thank you for your kind suggestion. Indeed, the glutamine metabolism is regulated by many factors, not just hormones. As you mentioned, this hormone-glutamine axis should not work for the triple-negative breast cancers, and glutamine may impact on TNBC through other ways independent of hormones. We have supplemented the discussion in this paragraph.